# Decreased Sensitivity of Rapid Antigen Test Is Associated with a Lower Viral Load of Omicron than Delta SARS-CoV-2 Variant

Théophile Cocherie,[a] Mathilda Bastide,[a] Soraya Sakhi,[a] Karen Zafilaza,[a] Philippe Flandre,[b] Valentin Leducq,[a] Aude Jary,[a] Sonia Burrel,[a] Martine Louet,[c] Vincent Calvez,[a] Anne-Geneviève Marcelin,[a] Stéphane Marot[a]

[a]INSERM, Institut Pierre Louis d'Epidémiologie et de Santé Publique, Assistance Publique-Hôpitaux de Paris (AP-HP), Hôpital Pitié-Salpêtrière, Service, Department of Virology, Sorbonne Université, Paris, France
[b]INSERM, Institut Pierre Louis d'Epidémiologie et de Santé Publique, Sorbonne Université, Paris, France
[c]Hôpital Pitié-Salpêtrière, Service de Santé au Travail, Assistance Publique-Hôpitaux de Paris, Sorbonne Université, Paris, France

**ABSTRACT** Large-scale screening for SARS-CoV-2 infection is an important tool for epidemic prevention and control. The appearance of new variants associated with specific mutations can call into question the effectiveness of rapid diagnostic tests (RDTs) deployed massively at national and international levels. We compared the clinical and virological characteristics of individuals infected by Delta or Omicron variants to assess which factors were associated with a reduced performance of RDT. A commercially available RDT as well as the evaluation of the viral load (VL) and the detection of replicate intermediates (RIs) were carried out retrospectively on positive SARS-CoV-2 nasopharyngeal specimens from health care workers of the Pitié-Salpêtrière Hospital infected by the Delta or Omicron variant between July 2021 and January 2022. Of the 205 samples analyzed (104 from individuals infected with Delta and 101 with Omicron), 176 were analyzed by RDT and 200 by RT-PCR for VL and RIs. The sensitivity of the TDR for Omicron was significantly lower than that observed for Delta (53.8% *versus* 74.7%, respectively, $P < 0.01$). Moreover, the Delta VL was significantly higher than that measured for Omicron (median Ct 21.2 *versus* 24.1, respectively, $P < 0.01$) and associated with the positivity of the RDT in multivariate analysis. We demonstrate a lower RDT sensitivity associated with a lower VL at the time of diagnosis on Omicron-infected individuals in comparison to those infected with the Delta variant. This RDT lower sensitivity should be taken into account in the large-scale screening strategy and in particular in case of strong suspicion of infection where testing should be repeated.

**IMPORTANCE** Previous reports have shown a variability in the diagnostic performance of RDTs. In the era of SARS-CoV-2 variants and the use of RDT, mutation associated with these variants could affect the test performance. We evaluate the sensitivity of the RDT Panbio COVID-19 Ag (Abbott) with two variants of concern (VOC), the Delta and Omicron variants. In order to investigate whether clinical characteristics or virological characteristics can affect this sensitivity, we collected clinical information and performed a specific RT-PCR that detected the RIs as a marker of the viral replication and viral cycle stage. Our results showed that Omicron was less detected than the Delta variant. A lower viral load of Omicron variant in comparison to Delta variant explained this decreased sensitivity, even if they are at the same stage of the disease and the viral cycle and should be taken into account with the use of RDT as diagnostic tool.

**KEYWORDS** COVID-19, SARS-CoV-2, antigen, diagnostics, viral load

In the context of the COVID-19 pandemic, large-scale screening for SARS-CoV-2 infection with fast reporting results are major tools of the control strategy for the COVID-19 pandemic and the aim of disrupting the chains of contamination (1).

Address correspondence to Stéphane Marot, stephanesylvain.marot@aphp.fr.

The authors declare no conflict of interest.

**TABLE 1** Clinical and virological characteristics of individuals infected by Delta or Omicron SARS-CoV-2 variants of concern[a]

| Variables | Overall (n) = 205 | | Delta variant (n = 104) | | Omicron variant (n = 101) | | |
|---|---|---|---|---|---|---|---|
| | n | | n | | n | | P-value |
| Male, n (%) | 205 | 62 (30.2) | 104 | 33 (31.7) | 101 | 29 (28.7) | 0.6 |
| Age (yrs), median [IQR] | 205 | 34 [27–48] | 104 | 36 [29–50] | 101 | 32 [26–41] | 0.016 |
| Positive RDT, n (%) | 176 | 112 (63.6) | 83 | 62 (74.7) | 93 | 50 (53.8) | 0.004 |
| Time between onset of symptoms and clinical sampling (days), median [IQR] | 118 | 1 [1–3] | 51 | 1 [1–2] | 67 | 1 [0–3] | 0.7 |
| Time between onset of symptoms and last vaccine injection (days), median [IQR] | 104 | 130 [25–201] | 45 | 167 [99–220] | 59 | 41 [18–159] | 0.001 |
| COVID-19: | 147 | | 66 | | 81 | | 0.7 |
| - Asymptomatic individuals, n (%) | | 53 (36.1) | | 26 (39.4) | | 27 (33.3) | |
| - Symptomatic individuals, n (%) | | 94 (63.9) | | 40 (60.6) | | 54 (66.7) | |
| No. of symptoms, median [IQR] | 145 | 2 [1–3] | 66 | 2 [1–2] | 79 | 2 [1–3] | 0.01 |
| Rhinitis, n (%) | 145 | 56 (38.6) | 66 | 29 (43.9) | 79 | 27 (34.2) | 0.2 |
| Fever, n (%) | 145 | 51 (35.2) | 66 | 18 (27.3) | 79 | 33 (41.8) | 0.069 |
| Dry cough, n (%) | 145 | 51 (35.2) | 66 | 16 (24.2) | 79 | 35 (44.3) | 0.012 |
| Headache, n (%) | 145 | 39 (26.9) | 66 | 14 (21.2) | 79 | 25 (31.6) | 0.2 |
| Sore throat, n (%) | 145 | 22 (15.2) | 66 | 2 (3) | 79 | 20 (25.3) | <0.001 |
| Asthenia, n (%) | 145 | 28 (19.3) | 66 | 10 (15.2) | 79 | 18 (22.8) | 0.2 |
| Anosmia, n (%) | 145 | 10 (6.9) | 66 | 9 (13.6) | 79 | 1 (1.3) | 0.006 |
| Ageusia, n (%) | 145 | 5 (3.5) | 66 | 5 (7.6) | 79 | 0 (0) | 0.018 |
| Myalgia, n (%) | 145 | 34 (23.5) | 66 | 11 (16.7) | 79 | 23 (29.1) | 0.078 |
| SARS-CoV-2 vaccine scheme | 142 | | 65 | | 77 | | <0.001 |
| - No vaccination | | 5 (3.5) | | 4 (6.2) | | 1 (1.3) | |
| - 1 injection | | 3 (2.1) | | 3 (4.6) | | 0 (0) | |
| - 2 injections | | 91 (64.1) | | 51 (78.5) | | 40 (51.9) | |
| - 3 injections | | 43 (30.3) | | 7 (10.8) | | 36 (46.8) | |
| SARS-CoV-2 E gene (Ct), median [IQR] | 200 | 23.1 [18.8–27.1] | 100 | 21.2 [17.8–25.2] | 100 | 24.1 [21.2–28.0] | <0.001 |
| Presence of RIs, n (%) | 200 | 174 (87.0) | 100 | 89 (89.0) | 100 | 85 (85.0) | 0.4 |

[a]Ct, cycle threshold; E gene, SARS-CoV-2 envelope glycoprotein gene; IQR, interquartile range; RDT, rapid diagnostic test; RI, replicative intermediate RNAs; RNA, RNA; SARS-CoV-2, severe acute respiratory syndrome coronavirus. Statistical comparisons were performed using Chi-squared test or Fisher's exact test for categorical variables and Mann-Whitney U-test for continuous variables.

To assist the use of standard RT-PCR tests carried out by laboratories, rapid diagnostic tests (RDT) based on antigen detection have been deployed on a massive scale at national and international levels. Sensitivity is a major criterion for screening tests to detect individuals infected with SARS-CoV-2 as fast as possible (2). RDTs were developed on the wild-type SARS-CoV-2 antigens, and since then, new variants of concern (VOCs) have been identified with specific patterns of mutations that could impact their detection due to the modification of epitopes. Recently, the emergence and rapid spread of the highly mutated Omicron variant, responsible for the fifth wave in the European Union (3), could impact a screening strategy based on RDTs in comparison to a previous circulating VOC Delta because of alteration of antigen recognition.

In this study, we aimed to estimate the sensitivity of a commercial RDT against these two major VOCs circulating during the fourth and fifth waves in France and to assess the factors associated with the performance of this RDT.

**Results.** In total, 205 participants were included, 104 individuals infected with Delta and 101 with Omicron VOC. We realized 200 RT-PCRs and 176 RDTs because of an insufficient available quantity of sample and obtained 145 participant questionnaires.

First, we compared the clinical and virological characteristics of individuals infected with the two VOCs. Participants were mostly female (68.3% versus 71.3%, P = 0.6), but Omicron-infected individuals were significantly younger than those infected with Delta (32 [26 to 41] versus 36 [29 to 50] years old respectively, P = 0.016). There was no difference in the delay between onset of symptoms and sampling for Delta and Omicron (1 [1 to 2] versus 1 [0 to 3] days, P = 0.7) but a significant higher delay between last vaccine injection and onset of symptoms for Delta variant (167 [99 to 220] versus 41 [1 to 159] days, P = 0.001) (Table 1). The mean number of symptoms was higher in the Omicron group (2 [1 to 3] versus 2 [1 to 2], P = 0.01) with a different pattern of clinical signs; Omicron-infected individuals had

**TABLE 2** Factors associated with a positive SARS-CoV-2 rapid diagnostic test (RDT)[a]

| Variables | Univariate analysis | | | Multivariate analysis | | |
|---|---|---|---|---|---|---|
| | OR | 95% CI | P value | OR | 95% CI | P value |
| SARS-CoV-2 Delta variant | 2.54 | 1.35–4.88 | 0.004 | 1.09 | 0.09–12.36 | 0.943 |
| Female sex | 1.22 | 0.62–2.36 | 0.556 | - | - | - |
| Time between onset of symptoms and clinical sampling | 0.86 | 0.71–1.04 | 0.127 | 0.88 | 0.58–1.27 | 0.504 |
| Time between onset of symptoms and last vaccine injection | 1.01 | 0.99–1.01 | 0.066 | - | - | - |
| No. of symptoms | 1.34 | 1.02–1.79 | 0.042 | 0.69 | 0.19–2.16 | 0.544 |
| Rhinitis | 0.77 | 0.36–1.61 | 0.485 | - | - | - |
| Fever | 2.20 | 1.01–5.10 | 0.055 | 1.18 | 0.09–17.76 | 0.898 |
| Dry cough | 1.95 | 0.91–4.33 | 0.092 | 5.59 | 0.62–78.24 | 0.146 |
| Headache | 0.73 | 0.33–1.63 | 0.443 | - | - | - |
| Sore throat | 0.63 | 0.25–1.63 | 0.337 | - | - | - |
| Asthenia | 1.02 | 0.42–2.63 | 0.959 | - | - | - |
| Anosmia | 2.99 | 0.46–58.14 | 0.325 | - | - | - |
| Ageusia | - | - | - | - | - | - |
| Myalgia | 3.17 | 1.26–9.15 | 0.020 | 5.87 | 0.58–109.4 | 0.166 |
| SARS-CoV-2 E gene (Ct) | 0.44 | 0.31–0.56 | <0.0001 | 0.34 | 0.14–0.55 | 0.001 |
| SARS-CoV-2 RIs (Ct) | 0.47 | 0.36–0.59 | <0.0001 | - | - | - |

[a]CI, confidence interval; Ct, cycle threshold; E gene, SARS-CoV-2 envelope glycoprotein gene; OR, odds ratio; RI, replicative intermediate RNAs; RNA, RNA; -, parameter not included in uni- or multivariate analysis.

significantly more dry cough and sore throat than Delta-infected individuals ($P = 0.012$ and $P < 0.001$, respectively), and they tended to present more fever and myalgia, whereas Delta-infected individuals had significantly more anosmia and ageusia symptoms ($P = 0.006$ and $P = 0.018$, respectively). The VL was significantly higher for Delta than for Omicron (Ct values: 21.2 [17.8 to 25.2] *versus* 24.1 [21.2 to 28.0], $P < 0.001$) and no significant difference was observed concerning the presence of RIs (89% *versus* 85% for Delta and Omicron, $P = 0.4$). The RDT's sensitivity for Omicron was significantly lower than that observed for Delta (53.8% versus 74.7%, $P = 0.004$) (Table 1).

Then, factors associated with the RDT's sensitivity were investigated. On univariate analysis, a better sensitivity was associated with Delta variant infection (odds ratio [OR] 2.54, 95% confidence interval [CI] 1.35 to 4.89, $P = 0.004$), number of symptoms (OR 1.34, 95%CI 1.02 to 1.79, $P = 0.042$), myalgia reporting (OR 3.17, 95%CI 1.26 to 9.15, $P = 0.020$) or a higher VL (OR 0.44, 95%CI 0.31 to 0.57, $P < 0.0001$) (Table 2). No significant association was found with the delay between onset of symptoms and sampling (OR 0.86, 95% CI 0.71 to 1.04, $P = 0.127$) or the delay between last vaccine injection and onset of symptoms (OR 1.01, 95% CI 0.99 to 1.01, $P = 0.066$). Interestingly, a high level of VL remained significantly associated with the positivity of the RDT (OR 0.34, 95% CI 0.14 to 0.55, $P = 0.001$) independently of the SARS-CoV-2 variant (Table 2).

**Discussion.** At the time of diagnosis, our results show that the RDT's sensitivity is decreased for Omicron variant compared to Delta variant, associated with a lower VL in nasopharyngeal samples of individuals rather than an alteration of the N-antigen recognition by the RDT. These results are in accordance with other studies that explored different available RDTs in patients infected by these two VOCs (4, 5). Moreover, a lower VL of the Omicron variant was also supported by two studies based on infectious or genomic VLs (6, 7).

When the fifth epidemic wave occurred in December 2021, we were concerned about the low VL of Omicron that could be the consequence of early diagnosis due to broad screening before the end of year gathering rather than a low viral replication. However, in our study, the delay between onset of symptoms and sampling was comparable between Omicron- and Delta-infected individuals and proportion of samples with RIs were similar, which allows us to argue either a similar infection stage of the disease and a similar replication cycle phase for both groups. A study demonstrated that Omicron VL was highest 2 to 5 days after diagnosis or after symptom onset (8).

This lower VL at the time of diagnosis could be explain by the characteristics of our population with a possible better immune status in Omicron-infected individuals.

Indeed, we observed a significant difference of the delay between last vaccine injection and onset of symptoms that was probably explained by the vaccine strategy in France: health care workers were required to receive a booster injection since early December 2021 concurrently to the 5th wave of infection.

We also showed a different pattern of symptoms between Omicron and Delta variants with more dry cough and sore throat for Omicron and more anosmia and ageusia for Delta. These results are in accordance with other studies carried out in asymptomatic and symptomatic individuals (9, 10), strengthening our findings.

Nevertheless, our study presents some limits. First, due to its retrospective design on frozen samples, although this conservation method should not affect antigen conformation and was identical for Delta and Omicron positive nasopharyngeal samples, limiting the bias of direct intervariant comparison. Second, due to missing clinical data limiting our interpretation of the RDTs performances on symptomatic or asymptomatic groups which were mostly explored in the context of the historical SARS-CoV-2 lineage (11, 12).

Number of studies have shown that RDTs have a lower sensitivity than RT-PCR assays on the historical SARS-CoV-2 lineage. In our knowledge, our study is a first to demonstrate a lower RDT's sensitivity associated with a lower VL at the time of diagnosis on Omicron-infected individuals in comparison to Delta-infected individuals. This should be considered in the large-scale screening strategy and in particular in case of strong suspicion of infection where testing should be repeated.

**Materials and methods.** A retrospective study was carried out on SARS-CoV-2 RT-PCR positive nasopharyngeal specimen from health care workers of the Pitié-Salpêtrière Hospital between July 2021 and January 2022. Panbio COVID-19 Ag RDT (Abbott) detecting SARS-CoV-2 N-antigen, viral load (VL) and presence of replication intermediates (RIs) using RT-PCR assays as previously described (13) were performed on the remaining part of samples stored at −80°C. The clinical data were obtained from the Occupational Health Department. All participants' samples and data were collected in the context of routine clinical care and our institutional review board approved this study.

Continuous variables were expressed as median and interquartile range (IQR) and discrete variables were expressed as numbers and percentages. Group comparison was performed using Chi-squared test or Fisher's test for categorical variables, and Mann–Whitney U-test for continuous variables. Univariate and multivariate logistic regression analyses were performed with GraphPad Prism 6.0 to identify associated factors with a positive RDT result. Relevant factors with a $P < 0.20$ on univariate analysis were included in the multivariate analysis.

**Data availability.** The data sets generated and analyzed are available from the corresponding author on reasonable request.

## ACKNOWLEDGMENTS

We express our deep gratitude to all the staff members of the Virology Department, the Occupational Health Department, and all of the participants of this study.

This work was supported by the Agence Nationale de la Recherche sur le SIDA et les Maladies Infectieuses Emergentes (ANRS-MIE), AC43 Medical Virology, EMERGEN Consortium and the SARS-CoV-2 Program of the Faculty of Medicine of Sorbonne Université.

S.M., A.-G.M., and V.C. initiated the study and coordinated all work carried out; T.C., M.B., S.S., V.L., and K.Z. conducted the experiments; M.L. provided clinical data about participants; T.C., A.J., P.F., S.B., and S.M. analyzed the data; T.C. and S.M. wrote the paper. All the authors read, corrected the manuscript, and approved the final version.

We have no conflicts of interest to declare.

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
