## [Reviewer comments · Microbiology Spectrum]

Microbiology Spectrum

Decreased sensitivity of rapid antigen test is associated with a lower viral load of Omicron than Delta SARS-CoV-2 variant

Théophile Cocherie, Mathilda Bastide, Soraya Sakhi, Karen Zafilaza, Philippe Flandre, Valentin Leducq, Aude Jary, Sonia Burrel, Martine Louet, Vincent Calvez, Anne-Geneviève Marcelin, and Stephane Marot

Corresponding Author(s): Stephane Marot, Hôpital Pitié-Salpêtrière, Assistance Publique - Hôpitaux de Paris, Sorbonne Université, INSERM, Institut Pierre Louis d'Epidémiologie et de Santé Publique (iPLESP)

Review Timeline:

Submission Date:	May 23, 2022
Editorial Decision:	August 4, 2022
Revision Received:	August 14, 2022
Accepted:	August 26, 2022

Editor: Meghan Starolis

Reviewer(s): The reviewers have opted to remain anonymous.

Transaction Report:

DOI: <https://doi.org/10.1128/spectrum.01922-22>

August 4, 2022

Dr. Stephane Marot

Hôpital Pitié-Salpêtrière, Assistance Publique - Hôpitaux de Paris, Sorbonne Université, INSERM, Institut Pierre Louis d'Epidémiologie et de Santé Publique (iPLESP)

Department of virology

Paris

France

Re: Spectrum01922-22 (Decreased sensitivity of rapid antigen test is associated with a lower viral load of Omicron than Delta SARS-CoV-2 variant)

Dear Dr. Stephane Marot:

Thank you for submitting your manuscript to Microbiology Spectrum. The review process is now complete, and the reviewers have asked to address some concerns before the manuscript can be considered for publication. When submitting the revised version of your paper, please provide (1) point-by-point responses to the issues raised by the reviewers as file type "Response to Reviewers," not in your cover letter, and (2) a PDF file that indicates the changes from the original submission (by highlighting or underlining the changes) as file type "Marked Up Manuscript - For Review Only". Please use this link to submit your revised manuscript - we strongly recommend that you submit your paper within the next 60 days or reach out to me. Detailed instructions on submitting your revised paper are below.

Link Not Available

Sincerely,

Meghan Starolis

Journals Department
Reviewer comments:

Reviewer #1 (Comments for the Author):

The authors assessed the factors that affect the reduced performance of RDT among individuals infected with the Delta and Omicron variants. The paper is well written, and I must commend the researchers for a good job done.

I have therefore attached minor comments for your attention

METHODS

Line 105: Replaced categorical variables with discrete variables

How was positive RDT defined in your study. Was it a binary or continuous variable and how was it coded? Also, the researchers should list the number of factors that were included in the logistic regression model here in the methods section and indicate how they were measured and coded.

Reviewer #2 (Comments for the Author):

This is a interesting manuscript where a reduction in sensitivity of rapid antigen test for SARS-CoV-2 Omicron variant compared to delta variant is reported. Moreover, a lower viral load associated to omicron variant would explain the reduction in sensitivity.

However, I have some comments to be address prior to publication:

1. The authors should make a comment in the discussion about a potential bias on the sampling. With a 100 samples of each variant, it could be possible that the lower viral load in omicron would be associated to this sampling and not to the variant itself.
2. Mean Ct values are really low either in delta or in omicron samples group, so the viral loads are high in general. So, the authors should consider in the discussion other reasons for the sensitivity reduction other than the viral load. I do not believe that a sensitivity of 54% for omicron variants is only associated to low viral loads when the average Ct is 24.
3. There is a clear difference in the number of vaccine shots between delta an omicron individuals. 46% on omicron individuals got 3 shots, and only 11% for delta. The authors should analyze and discuss how this fact may explain the results reported. A stronger immunization could either cause a lower viral load or direct scavenge viral antigens.
4. If possible, the authors should also include previous infection as a variable. Probably, there are reinfected individuals in the omicron group and that could explain either the reduced viral load or the reduced sensitivity of the test for the same reasons discussed in point 3.

Staff Comments:

Preparing Revision Guidelines

Please return the manuscript within 60 days; if you cannot complete the modification within this time period, please contact me. If you do not wish to modify the manuscript and prefer to submit it to another journal, please notify me of your decision immediately so that the manuscript may be formally withdrawn from consideration by Microbiology Spectrum.

The authors assessed the factors that affect the reduced performance of RDT among individuals infected with the Delta and Omicron variants. The paper is well written, and I must commend the researchers for a good job done.

I have therefore attached minor comments for your attention

METHODS

Line 105: Replaced categorical variables with discrete variables

How was positive RDT defined in your study. Was it a binary or continuous variable and how was it coded? Also, the researchers should list the number of factors that were included in the logistic regression model here in the methods section and indicate how they were measured and coded.

Institut Pierre Louis d'Épidémiologie et de Santé Publique
Pierre Louis Institute of Epidemiology and Public Health

Unité mixte de recherche en santé n° 1136 (UMR-S 1136)
Directrice : Dominique Costagliola

Editorial Office

Microbiology Spectrum

Paris, 14th August 2022

Reviewer #1

The authors assessed the factors that affect the reduced performance of RDT among individuals infected with the Delta and Omicron variants. The paper is well written, and I must commend the researchers for a good job done.

I have therefore attached minor comments for your attention

We would like to thank the reviewer for his kind review and comments.

1. Line 105: Replaced categorical variables with discrete variables

As suggested by the reviewer we have modified the line 105.

2. How was positive RDT defined in your study? Was it a binary or continuous variable and how was it coded?

In this study, positive RDT was defined according to the manufacturer's recommendations, with the appearance of bands, even a simple trace, at two specific heights on the RDT: one marking SARS-CoV-2 antigen positivity and the other marking successful completion of the technique. It was a binary variable coded "negative" or "positive".

3. Also, the researchers should list the number of factors that were included in the logistic regression model here in the methods section and indicate how they were measured and coded.

We fully agree with the reviewer that adding informations of the variables included in the logistic regression model could improve our paper. Variables included in the multivariate analysis appeared in the Table 2 and we have added in the methods section how these variables were measured and coded (lines 110 to 115).

Reviewer #2 (Comments for the Author):

This is an interesting manuscript where a reduction in sensitivity of rapid antigen test for SARS-CoV-2 Omicron variant compared to delta variant is reported. Moreover, a lower viral load associated to omicron variant would explain the reduction in sensitivity.

However, I have some comments to be addressed prior to publication:

We would like to thank the reviewer for his kind review and comments.

1. The authors should make a comment in the discussion about a potential bias on the sampling. With a 100 samples of each variant, it could be possible that the lower viral load in omicron would be associated to this sampling and not to the variant itself.

We agree with the reviewer about this potential bias. As suggested, we have added a comment about this limitation in the discussion section (line 177).

2. Mean Ct values are really low either in delta or in omicron samples group, so the viral loads are high in general. So, the authors should consider in the discussion other reasons for the sensitivity reduction other than the viral load. I do not believe that a sensitivity of 54% for omicron variants is only associated to low viral loads when the average Ct is 24.

We also agree about the questioning reduce sensitivity for Omicron variant even with an estimated high viral load. Nevertheless, we observed in our study, in absence of difference on the delay between the symptoms onset and sampling, a shift of 3 Ct (median Ct of 21.2 for Delta variant and 24.1 for Omicron variant), which lead to a potential 10-fold decrease of the viral load for Omicron variant in comparison to Delta variant, which could explain partially this reduce sensitivity. We also discuss this point in the other reviewer's remarks.

3. There is a clear difference in the number of vaccine shots between delta and omicron individuals. 46% on omicron individuals got 3 shots and only 11% for delta. The authors should analyze and discuss how this fact may explain the results reported. A stronger immunization could either cause a lower viral load or directly scavenge viral antigens.

We thank the reviewer about this particular point. Indeed, the vaccine status differ between healthcare workers (HCW) infected by Omicron or Delta variant, because of the time of infection and the requirement of a third dose for the HCW during autumn 2021. We initially do not develop this part in the manuscript, except lines 164 to 165, due to the number of words allowed but we are totally agree about the possibility of a lower VL of Omicron explained by a stronger immunization and/or a direct scavenging of viral antigens. We have added a comment for this point in the manuscript (lines 165-166). However, we do not find significant difference for RDT positivity concerning the schema of vaccination (3 dose versus 2 dose or less: OR 0.59, 95%CI 0.27-1.2, p=0.17) but this absence of significance could be explained by the size of the sampling.

4. If possible, the authors should also include previous infection as a variable. Probably, there are reinfected individuals in the omicron group and that could explain either the reduced viral load or the reduced sensitivity of the test for the same reasons discussed in point 3.

Unfortunately, we do not have this information for the majority of the participants and to avoid any declarative bias (as remembering or asymptomatic infection), we have decided to not include this variable. However, we have added this point as a limit of the study (line 179).

August 26, 2022

Dr. Stephane Marot
Hôpital Pitié-Salpêtrière, Assistance Publique - Hôpitaux de Paris, Sorbonne Université, INSERM, Institut Pierre Louis
d'Epidémiologie et de Santé Publique (iPLESP)
Department of virology
Paris
France

Re: Spectrum01922-22R1 (Decreased sensitivity of rapid antigen test is associated with a lower viral load of Omicron than Delta SARS-CoV-2 variant)

Dear Dr. Stephane Marot:

Your manuscript has been accepted as I feel you have adequately addressed the comments and concerns of the reviewers. I am forwarding it to the ASM Journals Department for publication. You will be notified when your proofs are ready to be viewed.

Sincerely,

Meghan Starolis
Editor, Microbiology Spectrum
